# Intranasal Administration of Cold-Adapted Live-Attenuated Eurasian Avian-like H1N1 Vaccine Candidate Confers Protection Against Different-Lineage H1N1 Viruses in Mice

**DOI:** 10.3390/vaccines13060596

**Published:** 2025-05-30

**Authors:** Qiu Zhong, Zuchen Song, Fei Meng, Yanwen Wang, Yijie Zhang, Zijian Feng, Yali Zhang, Yujia Zhai, Yan Chen, Chuanling Qiao, Hualan Chen, Huanliang Yang

**Affiliations:** State Key Laboratory for Animal Disease Control and Prevention, Harbin Veterinary Research Institute, Chinese Academy of Agricultural Sciences, Harbin 150069, China

**Keywords:** Eurasian avian-like H1N1 swine influenza virus, cold-adapted vaccine, intranasal administration, protective efficacy

## Abstract

Background/Objectives: Eurasian avian-like (EA) H1N1 swine influenza viruses, with their persistent evolution and zoonotic potential, seriously threaten both swine and human health. The objective was to develop an effective vaccine against these viruses. Methods: A cold-adapted, temperature-sensitive live-attenuated influenza vaccine (LAIV) candidate, GX18*ca*, was developed. It was derived from the wild-type EA H1N1 strain A/swine/Guangxi/18/2011 (GX18) through serial passaging in embryonated eggs at temperatures decreasing from 33 °C to 25 °C. Its characteristics were studied in mice, including attenuation, immune responses (mucosal IgA, serum IgG, IFN-γ+ CD4^+^/CD8^+^ T-cell responses), and protective efficacy against homologous (GX18), heterologous EA H1N1 (LN972), and human 2009/H1N1 (SC1) viruses. Results: GX18*ca* showed cold-adapted and temperature-sensitive phenotypes. In mice, it was attenuated, with viral titers in the nasal turbinates and lungs reduced 1000–10,000-fold compared to the wild-type strain, and it cleared by day 5 post infection. Intranasal immunization elicited strong cross-reactive immune responses. Mucosal IgA had broad reactivity, and serum IgG titers reached high levels. IFN-γ+ CD4^+/^CD8^+^ T-cell responses were detected against all the tested viruses. A single dose of GX18*ca* fully protected against GX18 and LN972 challenges, and two doses significantly reduced SC1 lung viral loads, preventing mortality and weight loss. Conclusions: GX18*ca* is a promising LAIV candidate. It can induce broad immunity, addressing the cross-protection gaps against evolving EA H1N1 SIVs and zoonotic H1N1 variants, which is crucial for swine influenza control and pandemic preparedness.

## 1. Introduction

Influenza A virus (IAV) evolves rapidly in nature and frequently jumps across species, constantly threatening both human and animal health. For humans, some animal-derived IAVs can cause transient infections, local endemic outbreaks, or even global pandemics. The H1N1 subtype, one of the earliest discovered influenza viruses, has inflicted substantial harm on both humans and animals. Eurasian avian-like (EA) H1N1 swine influenza viruses (SIVs) have become the dominant subtype in pig populations across China, endangering both animal and human health. Since the introduction of the 2009 pandemic H1N1 (2009/H1N1) virus into swine populations, reassortant viruses between 2009/H1N1 and endemic swine viruses have been detected in pigs in many countries [1,2,3,4].

Previous studies indicate that novel EA H1N1 reassortant strains have replaced early EA H1N1 strains and emerged as dominant ones in Chinese pig populations. These strains exhibit higher pathogenicity and transmission capacity among mammals. They can infect humans, sometimes even resulting in death. The EA H1N1 virus is found to be genetically and antigenically distinct from the 2009/H1N1 viruses that have circulated over the last 15 years. Antibodies generated in the human population as a result of infection and vaccination are not sufficient to defend against the attack of this type of virus [5,6,7].

Vaccination represents the most cost-effective strategy for preventing influenza infections. Currently, inactivated vaccines are used for swine influenza prevention and control in many countries. However, their limited capacity to induce mucosal and cellular immunity has raised concerns about their effectiveness in providing broad-spectrum and long-term protection. Live-attenuated influenza vaccines (LAIVs) present a promising alternative [8,9,10]. They mimic natural infections and elicit robust immune responses, including mucosal immunity, serum antibodies, and T cell-mediated immunity. Research on LAIVs in swine have repeatedly shown their good safety and higher efficacy than inactivated vaccines under experimental conditions mimicking field settings [11]. Cold-adapted (*ca*) LAIVs, restricted in replication at higher temperatures, enhance safety by confining viral replication to the upper respiratory tract. This not only reduces the risk of vaccine-associated adverse effects but also boosts immunogenicity by stimulating localized immune responses.

In this study, we developed a cold-adapted EA H1N1 SIV vaccine candidate (GX18*ca*) derived from the wild-type strain A/swine/Guangxi/18/2011 (GX18). We evaluated its safety, immunogenicity, and cross-protective efficacy in a mouse model. Our findings demonstrate that intranasal administration of GX18*ca* induces robust humoral, mucosal, and cellular immune responses, conferring protection against homologous and heterologous influenza virus challenges, including the human 2009/H1N1 strain.

## 2. Materials and Methods

### 2.1. Viruses and Cells

A/swine/Guangxi/18/2011 (GX18, GenBank accession no. KP404217-KP404224), an EA H1N1 influenza virus, was used for cold adaptation. For challenge studies and related investigations, the homologous GX18 virus, along with two heterologous influenza viruses from the same or different hemagglutinin (HA) lineages, was selected according to previous studies [5,6]. These heterologous viruses included A/swine/Liaoning/972/2016 (LN972), a reassortant EA H1N1 virus, and A/Sichuan/1/2009 (SC1), a human 2009/H1N1 virus. All viruses were propagated in 10-day-old specific-pathogen-free (SPF) embryonated chicken eggs. Madin–Darby canine kidney (MDCK) cells were cultured in Dulbecco’s Modified Eagle Medium (DMEM) supplemented with 10% fetal bovine serum.

### 2.2. Cold Adaptation of H1N1 Influenza Virus

The cold-adapted vaccine strain GX18*ca* was generated through serial passage of the wild-type EA H1N1 SIV strain GX18 in SPF embryonated chicken eggs. The virus was initially propagated at 33 °C and gradually adapted to lower temperatures; the virus was passaged more than once at each temperature from 33 °C to 26 °C and five times at 25 °C. For each passage, five eggs were inoculated with four HA units (HAUs) (0.2 mL) of the virus diluted in PBS (phosphate-buffered saline, pH 7.4), then incubated for 72–96 h.

### 2.3. Cold Adaptation and Temperature Sensitivity Phenotype Analysis of the GX18ca Virus

We evaluated the *ca* phenotype, a temperature-sensitive (*ts*) phenotype of the GX18*ca* virus as previously described [12]. The *ca* phenotype is defined as a less than 100-fold reduction in the virus titer in eggs at 25 °C compared with that at 33 °C. The *ts* phenotype is defined as a more than 100-fold reduction in the virus titer in eggs at 39 °C compared with that at 37 °C. The strains GX18 and GX18*ca* were diluted from 10^−2^ to 10^−9^ using phosphate-buffered saline (PBS). Four 10-day-old SPF chicken embryos were inoculated with the diluted viral solutions at each dilution. These embryos were then incubated for 72 h at 25 °C, 33 °C, 37 °C, and 39 °C, respectively. After incubation, the allantoic fluid of each chicken embryo was collected, and the HA titers were determined with 0.5% chicken red blood cells. The Reed–Muench method was employed to calculate the 50% embryo infectious dose (EID_50_).

### 2.4. Multiple-Step Growth Kinetics of GX18ca and GX18 in MDCK Cells

To assess the in vitro replication capabilities of GX18*ca* and GX18 viruses, confluent monolayer cultures of MDCK cells were infected in triplicate, with each virus at a multiplicity of infection (MOI) of 0.01. Post infection, the cells were incubated in Opti-MEM (Gibco, Grand Island, NY, USA) supplemented with 0.2 g/mL of L-(tosylamido-2-phenyl) ethyl chloromethyl ketone (TPCK)-treated trypsin (Sigma-Aldrich, St. Louis, MO, USA). The incubation was carried out at one of four temperatures: 30 °C, 33 °C, 37 °C, or 39 °C. At 12, 24, 48, and 72 h post inoculation (p.i.), cell culture supernatants were collected. Virus titers within these supernatants were determined using MDCK cells, and each experiment was repeated three times, following the protocol described previously [13].

### 2.5. Replication of H1N1 Influenza Viruses GX18ca and GX18 in Mice

To test the attenuation phenotype of the viruses, groups of 20 six-week-old SPF female BALB/c mice (procured from Beijing Merial Vital Laboratory Animal Technology Co. Ltd., Beijing, China) were lightly anesthetized with CO_2_. Notably, CO_2_ anesthesia, as an uncommon procedure that may impose additional stress on mice, could have influenced the survival outcome of the PBS control group. They were then infected intranasally (i.n.) with 10^6^ EID_50_/mouse of the H1N1 GX18*ca* or GX18 virus or with 50 µL of PBS as a control. Three mice per group were euthanized on days 1, 2, 3, 4, and 5 p.i. The nasal turbinates, brain, lungs, spleen, and kidneys were collected as tissue samples for virus titration in eggs. Each 0.1 g tissue sample was homogenized in 1 mL of PBS (pH 7.4), and viral titers in the resulting homogenates were determined and expressed as log_10_ EID_50_/mL. The remaining five mice in each group were observed daily for body weight changes and mortality. All animal experiments were carried out in strict accordance with the recommendations in the *Guide for the Care and Use of Laboratory Animals of the Ministry of Science and Technology of the People’s Republic of China*. All studies were conducted in a biosecurity level 2 laboratory approved for such use by the Harbin Veterinary Research Institute (HVRI) of the Chinese Academy of Agricultural Sciences (CAAS). The protocol was approved by the Committee on the Ethics of Animal Experiments of the HVRI of the CAAS on 6 April 2018 (approval number: SY-2018-mi-108).

### 2.6. Evaluation of Immunogenicity of the GX18ca Vaccine Candidate in Mice

Fifty six-week-old female BALB/c mice were randomly divided into two groups (20 and 30 mice, respectively). The 20-mouse group received a single intranasal immunization with 50 μL of GX18*ca* at a viral dose of 10^6^ EID_50_/mouse, while the 30-mouse group underwent a double-immunization regimen at a 3-week interval. To assess cellular immune responses induced by the GX18*ca* vaccine candidate, five mice were randomly euthanized 1 week post immunization for splenocyte preparation, as described previously [14]. Splenocytes were isolated, mechanically dissociated, and filtered to remove debris. After lysing erythrocytes, viable lymphocytes (2 × 10^7^ cells/mL) were stimulated with 0.1 MOI of GX18, LN972, or SC1 viruses. The stimulation occurred at 37 °C in a 5% CO_2_ incubator for 6 h. Brefeldin A (10 μg/mL) was added during the final 4 h to block cytokine secretion. First, cells were stained with fluorescent-conjugated anti-CD4 (FITC) and anti-CD8 (PE) antibodies. Then, they were fixed, permeabilized, and intracellularly stained for IFN-γ (APC). Samples were acquired on a BD LSRFortessa flow cytometer and analyzed with FlowJo software 10.8.1, excluding dead cells (via 7-AAD staining). IFN-γ + frequencies in CD4^+^ and CD8^+^ T cells were calculated, and data were presented as mean ± SD (*n* = 5). In the single-immunization group, serum samples were obtained from three randomly selected mice at weeks 2, 4, 6, and 8 post immunization. These samples were used for IgG antibody quantification. In the double-immunization group, blood was collected via retro-orbital bleeding from three randomly selected mice at weeks 4, 5, and 6 after the initial immunization. The serum isolated from these blood samples was used for quantification of hemagglutination inhibition (HI) antibodies and neutralizing antibodies. Serum samples collected at weeks 4 and 6 after the initial immunization were also used for IgG antibody detection. Moreover, nasal washes and bronchoalveolar lavage fluids (BALFs) were collected from three mice in the double-immunization group at weeks 4 and 6 post initial immunization. These samples were utilized to evaluate secretory IgA (SIgA) antibody levels.

### 2.7. Evaluation of the Efficacy of the GX18ca Virus Vaccine Candidate in Mice

Groups of six-week-old female BALB/c mice were i.n. immunized with either one or two doses of 10^6^ EID_50_ of GX18*ca* or PBS (mock-immunized) at a 3-week interval. Three weeks after single immunization and three weeks after double immunization, groups of eight mice were challenged i.n. with 50 μL of a solution containing 10^6^ EID_50_ of the GX18, LN972, or SC1 viruses. At 3 days post-challenge, the nasal turbinates, lungs, kidneys, spleen, and brain were harvested. Tissue homogenates were prepared as previously described and titrated in eggs. Virus titers in each organ were expressed as log_10_ EID_50_/mL of tissue from three mice per group. The remaining mice were monitored daily and weighed for 14 days post-challenge.

### 2.8. Statistical Analysis

GraphPad Prism software (version 8.00) was used for all statistical calculations. Data were first examined for normality using Shapiro–Wilk’s tests.

## 3. Results

### 3.1. Generation of a Cold-Adapted and Temperature-Sensitive H1N1 Swine Influenza Virus GX18ca

The H1N1 cold-adapted GX18*ca* was generated by serial passages of GX18 in SPF embryonated eggs at successively lower temperatures from 33 °C to 25 °C and then purified five times by limiting dilution in SPF embryonated chicken eggs at 25 °C. To determine whether the cold-adapted H1N1 vaccine candidate exhibited the *ca* phenotype, GX18*ca* or GX18 was inoculated in SPF embryonated chicken eggs at 25 °C or 33 °C for 72 h, after which the virus titers were measured. As shown in Figure 1A, GX18*ca* grew to 7.5 ± 0.2 log_10_ EID_50_/mL and 6.7 ± 0.3 log_10_ EID_50_/mL in eggs at 33 °C and 25 °C, respectively, and the difference in the replication titers of the GX18*ca* strain between the two growth conditions was within 100-fold; GX18*ca* grew to 7.5 ± 0.2 log_10_ EID_50_/mL in eggs at 37 °C, while GX18*ca* was not able to grow at 39 °C, and the difference in the growth titer between 37 °C and 39 °C was more than 100-fold. These results indicated that GX18*ca* exhibited the *ca* and *ts* phenotype.

The multiple-step growth kinetics of GX18*ca* and GX18 in MDCK cells revealed that GX18 grew best at 33 °C for 48 h, with a viral titer of 7.3 ± 0.5 log_10_ TCID_50_/mL. GX18*ca* grew best at 30 °C for 72 h, with a viral titer of 7.7 ± 0.1 log_10_ TCID_50_/mL. The viral titer of GX18 cultured at 39 °C for 72 h was 5.3 ± 0.4 log_10_ TCID_50_/mL, while the GX18*ca* strain did not replicate at 39 °C. Compared with the parental strain, GX18*ca* exhibited the *ca* and *ts* phenotype in MDCK cells as well (Figure 1B,C).

### 3.2. Attenuation Confirmation of the Live-Attenuated GX18ca Vaccine Candidate in Mice

To assess the attenuated (*att*) phenotype of the GX18*ca* virus, we evaluated its replication in the upper and lower respiratory tracts of mice. Virus isolation was performed on the nasal turbinates and lungs of mice inoculated with the parental GX18 virus. From 1 to 5 days p.i., virus titers in the nasal turbinates of these mice ranged from 0.67 to 5.67 log_10_ EID_50_/mL. Simultaneously, in the lungs, virus titers ranged from 1.5 to 5.0 log_10_ EID_50_/mL (Figure 2A). In contrast, in the group inoculated with the GX18*ca* virus, virus isolation yielded low titers. Between 1 and 4 days p.i., mean virus titers in the nasal turbinates ranged from 0.67 to 2.67 log_10_ EID_50_/mL. In the lungs, titers ranged from 1 to 2.5 log_10_ EID_50_/mL. Significantly, no virus was isolated from the lungs or nasal turbinates of any mice on day 5 p.i. (Figure 2B). We also monitored body weight changes in infected mice. Mice infected with the GX18*ca* virus exhibited a growth trend similar to that of the PBS control group. Conversely, mice infected with the GX18 virus experienced a maximum average body weight loss of approximately 19.8% on day 7 p.i. (Figure 2C). No virus was recovered from the spleens, kidneys, and brains of any of the mice challenged with the viruses. Collectively, these results indicate that the GX18*ca* virus exhibits the *att* phenotype.

### 3.3. GX18ca Vaccine Candidate Elicits Anti-H1 Influenza Virus Cellular Immunity in Mice

Live-attenuated vaccines are known for their ability to effectively stimulate both cellular and humoral immune responses, often achieving this with a single dose. To evaluate the cellular immune efficacy of the GX18*ca* vaccine candidate, we investigated the CD4^+^ and CD8^+^ T-cell responses in mice one week after single immunization with the GX18*ca* virus. PBS-inoculated mice served as the control group, following the experimental protocol described previously [14].

After in vitro stimulation with the GX18, LN972, and SC1 viruses, IFN-γ+CD4^+^ T cells constituted 0.73%, 0.70%, and 0.67% of the splenic lymphocytes in immunized mice, respectively (Figure 3A). Compared to the PBS control group, mice immunized with the GX18*ca* vaccine candidate exhibited significantly enhanced CD4^+^ T-cell responses against all three viruses. Similarly, IFN-γ+CD8^+^ T cells accounted for 0.77%, 0.73%, and 0.77% of the splenic lymphocytes in immunized mice upon stimulation with the GX18, LN972, and SC1 viruses (Figure 3B). Mice immunized with the GX18*ca* vaccine candidate also demonstrated significant CD8^+^ T-cell activation in response to these viruses when compared to the control group. Collectively, these results indicate that the GX18*ca* strain can induce virus-specific T cells in mice. These T cells are capable of recognizing both the EA H1N1 and 2009/H1N1 influenza viruses. Evidently, the GX18*ca* vaccine candidate is able to trigger robust cellular immune responses against H1 subtype influenza viruses from different lineages in mice.

### 3.4. GX18ca Vaccine Candidate Elicits Cross-Reactive Mucosal SIgA Against H1 Influenza Viruses

SIgA specific to influenza viruses plays a crucial role in blocking the initial infection of pathogens. To further explore whether the GX18*ca* vaccine candidate can induce the production of antibodies that specifically bind to the GX18, LN972, and SC1 influenza viruses, we collected nasal washings and bronchoalveolar lavage fluids (BALFs) from mice at 4 and 6 weeks after the initial immunization. The levels of SIgA antibodies in these samples were determined by enzyme-linked immunosorbent assay (ELISA). SIgA antibodies specific to GX18, LN972, and SC1 were detected in both the nasal washings and the BALF samples. Mice immunized with GX18*ca* produced significantly higher levels of SIgA antibodies at 6 weeks post immunization compared to 4 weeks post immunization. Moreover, the SIgA levels in the nasal wash and the BALFs demonstrated broad cross-reactivity, suggesting that the GX18*ca* vaccine candidate can elicit a mucosal immune response capable of recognizing multiple H1 influenza viruses (Figure 4).

### 3.5. HI and NT Antibody Titers Induced by GX18ca Vaccine Candidate in Mice

To determine whether mice immunized with GX18*ca* could generate HI and NT antibodies against the GX18, LN972, and SC1 viruses, we collected double-immunization group mouse sera 4, 5, and 6 weeks after the initial immunization for HI and NT antibody assays as described previously [15]. After immunization, the mice produced relatively high levels of HI and NT antibodies against the wild-type parental virus GX18. These antibody titers peaked in the second week after the double immunization, reaching 1:213.3 for HI antibodies and 1:160 for NT antibodies. The titers of the HI and NT antibodies induced by the vaccine against LN972 were lower than those against GX18. The HI antibodies reached their highest level of 1:160 in the second week after the double immunization, while NT antibodies peaked at 1:111.7 in the third week. However, the titers of HI and NT antibodies against SC1 were the lowest among the three viruses. One, two, and three weeks after the double immunization, these titers remained below 1:50. Overall, these results suggest that the GX18*ca* vaccine candidate elicits higher titers of HI and NT antibodies against EA H1N1 viruses but lower titers against 2009/H1N1 viruses (Figure 5).

### 3.6. IgG Antibody Responses Induced by GX18ca Vaccine Candidate in Mice

To comprehensively unravel the humoral immune response of the host against specific antigens, the study determined IgG antibody titers in the sera of immunized mice at multiple time points post immunization. Serum samples (*n* = 3) were collected randomly from the single-immunization group at 2, 4, 6, and 8 weeks post immunization. For the double-immunization group, serum samples (*n* = 3) were collected randomly at 4 and 6 weeks after the initial immunization. The ELISA technique was utilized to measure IgG antibody levels in the collected samples.

Experimental results demonstrated that the GX18*ca* vaccine could elicit high-titer antibody production against the parental strain GX18. Specifically, antibody titers exceeded 1:100,000 at 8 weeks after the first immunization as well as at 1 and 3 weeks after double immunization. Moreover, the vaccine could induce the production of serum IgG antibodies against the LN972 and SC1 viruses. After single-dose immunization, serum IgG antibody titers against LN972 and SC1 were comparable, neither exceeding 1:20,000. However, after double immunization, the titer of IgG antibodies against LN972 was higher than that against SC1. Nevertheless, neither titer exceeded 1:40,000. Evidently, there was a substantial difference in IgG antibody titers, with double immunization leading to much higher levels than single immunization (Figure 6).

### 3.7. Protective Efficacy Against Homologous and Heterologous Challenges

To comprehensively evaluate the protective efficacy of the GX18*ca* vaccine candidate, we carried out a series of challenge experiments. Mice were pre-immunized with one or two doses of the GX18*ca* vaccine candidate and then challenged with the GX18, LN972, and SC1 viruses. After being challenged with the GX18 and LN972 viruses, no virus could be detected in the nasal turbinates and lung tissues of mice that had received either one or two doses of the vaccine (Figure 7A,B). The immunized mice demonstrated consistent weight gain. In sharp contrast, mock-immunized mice experienced a significant decline in body weight within one week post-challenge. Viruses were detected in the nasal turbinates and lungs of mock-immunized mice, with mean titers ranging from 6.7 to 2.5 log_10_ EID_50_/mL (Figure 7A,B). Mice immunized with a single dose of PBS lost approximately 14% of their average body mass after the GX18 challenge (Figure 7D). Similarly, all mice immunized with a single dose of PBS succumbed within nine days after the LN972 challenge (Figure 7H). These results clearly indicate that a single immunization with GX18*ca* provides complete protection against its parental wild-type virus GX18 and the reassortant EA H1N1 virus LN972.

Three weeks after the final immunization, mice were challenged with the SC1 virus at a dose of 10^6^ EID_50_. Three days post-challenge, nasal turbinate and lung viral titers in the single-PBS-immunization group were 5.7 ± 0.5 log_10_ EID_50_/mL and 7.3 ± 0.3 log_10_ EID_50_/mL, respectively. In the double-PBS-immunization group, these titers were 5.4 ± 0.8 log_10_ EID_50_/mL and 6.9 ± 0.3 log_10_ EID_50_/mL (Figure 7C). After the challenge, immunized mice continued to gain weight steadily. In contrast, mice in the mock-immunized groups suffered severe weight loss. Only 20% and 80% of the single-PBS-immunization and double-PBS-immunization groups survived, respectively (Figure 7F,I). In the single-immunization group, virus replication in the lungs post-challenge reached a titer of 4.5 log_10_ EID_50_/mL, approximately 600-fold lower than that in the mock-immunized mice. In the double-immunization group, virus replication in the lungs post-challenge reached a titer of 4.3 log_10_ EID_50_/mL, approximately 400-fold lower than that in the mock-immunized mice. Additionally, in the double-immunization group, the virus was completely cleared from the nasal turbinates of immunized mice 3 days post-challenge (Figure 7C). No virus was recovered from the spleens, kidneys, and brains of any of the mice challenged with the viruses. The results of these experiments strongly suggest that GX18*ca* vaccination confers protection against the 2009/H1N1 virus in mice. This is clearly evidenced by the prevention of death, the weight loss, and the significant reduction in viral loads.

## 4. Discussion

The persistent circulation of EA H1N1 SIVs within Chinese pig populations has established an antigenically evolving viral reservoir with zoonotic potential [5,6,16,17,18]. Although commercial inactivated swine influenza vaccines have been utilized in Chinese pig herds, their inability to induce mucosal immunity and their limited cross-reactivity against antigenic drift variants highlight the urgent need for vaccines capable of providing broad protection against evolving swine influenza viruses [19].

Live-attenuated influenza vaccines (LAIVs) possess distinct advantages. They can mimic natural infection processes, simultaneously triggering both humoral and cellular immune responses [20]. The cold-adaptation strategy, which has been successfully implemented in human influenza vaccines, achieves biological containment through *ts* replication while preserving immunogenicity. Intranasal cold-adapted live-attenuated influenza vaccines have been in use for over a decade and have proven highly effective in preventing seasonal influenza, especially among young children [21]. When constructing human LAIV strains, six internal genes from the master donor virus endow the vaccine with *ca*, *ts*, and *att* properties. Meanwhile, the two surface genes, HA and neuraminidase (NA), derived from a wild-type virus, confer antigenicity. The resulting vaccine is a reassortant virus based on the genetic background of a pre-established cold-adapted donor strain. However, if the same construction strategy is applied to swine influenza vaccine strains, genes from human influenza virus strains will be introduced into pig herds, posing uncertain biosafety risks.

Based on the aforementioned principles, we developed GX18*ca*, a cold-adapted EA H1N1 vaccine candidate, and systematically evaluated its safety, immunogenicity, and cross-protective efficacy in a mouse model. Infecting i.n. 6-week-old mice with GX18*ca* at a viral dose of 10^6^ EID_50_ per mouse did not result in mortality, indicating its low pathogenicity. The cold-adapted variant GX18*ca*, obtained through cold passage, exhibited further reduced virulence. This was manifested by lower viral titers in the nasal turbinates and lungs of mice and a shorter viral clearance time p.i., thus confirming its favorable safety profile. More importantly, the GX18*ca* strain belongs to the G1 genotype [6]. All of its eight gene segments are derived from the earliest EA H1N1 strains. Currently, strains of this genotype have disappeared from pig populations in China, where they have been replaced by the G4 and G5 genotypes [6,18]. Therefore, the application of the GX18*ca* strain will not elevate the risk of genetic reassortment with the currently circulating endemic strains, which could lead to the emergence of new genotype strains. Moreover, there is no likelihood that the GX18*ca* strain will regain its virulence within pig populations and evolve into a new prevalent strain. Our research findings demonstrate that GX18*ca* is safe, immunogenic, and capable of providing cross-protection against heterologous influenza viruses, including the pandemic 2009/H1N1 strain. Notably, in the post-immunization challenge test, when exposed to the same viral dose, mice in the double immunized PBS control group exhibited less severe pathogenicity, as evidenced by reduced body weight loss and mortality rates, than those with single immunization. This phenomenon is speculated to result from the increased age of the mice [22]. The absence of HI antibody cross-reactivity between GX18*ca* and the SC1 virus underscores the significance of non-neutralizing antibodies and T cell-mediated immunity in heterologous protection. This discovery aligns with previous studies on cold-adapted human influenza vaccines. These studies have shown that LAIVs can induce extensive cross-reactive immune responses [23,24]. The robust mucosal and cellular immune responses elicited by GX18*ca* highlight its potential to overcome the limitations of inactivated vaccines, which primarily induce humoral immunity.

## 5. Conclusions

GX18*ca* is a promising LAIV candidate. It can induce broad immunity, addressing the cross-protection gaps against evolving EA H1N1 SIVs and zoonotic H1N1 variants in mouse models. Given its characteristics, GX18*ca* may offer advantages in the rapid development and deployment of cold-adapted influenza vaccines during pandemics compared to traditional vaccines. Future research should focus on evaluating its efficacy in pigs and exploring its potential application in humans. Considering the continuous evolution of influenza viruses and the persistent threat of new pandemics, the research, development, and application of LAIVs such as GX18*ca* are essential for strengthening global influenza prevention and control strategies.

## Figures and Tables

**Figure 1 vaccines-13-00596-f001:**
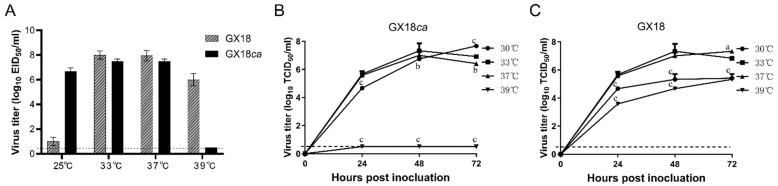
Replication of the GX18*ca* and GX18 virus under different temperatures. (**A**) The viral titers were determined by endpoint titration in eggs and expressed as mean log_10_ EID_50_/mL ± standard deviation (SD). Multi-step growth kinetics of GX18ca (**B**) and GX18 (**C**) at 30 °C, 33 °C, 37 °C, and 39 °C. MDCK cells were inoculated with GX18ca and GX18 at an MOI of 0.01 at 30 °C, 33 °C, 37 °C, and 39 °C. Infected cells were harvested at the indicated time points, and virus titers are expressed as the mean ± standard deviation (S.D.) log_10_ TCID_50_/mL. The growth data shown are the mean results of three independent experiments. At each time point, data were analyzed by one-way ANOVA followed by Dennett’s test by comparing the data of each group with values at 33 °C (a, *p* < 0.05; b, *p* < 0.01; c, *p* < 0.001). The dashed line indicates the limit of detection.

**Figure 2 vaccines-13-00596-f002:**
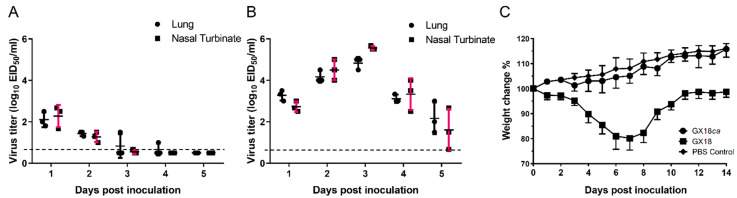
Comparative replication kinetics and pathogenicity of GX18*ca* and GX18 viruses in BALB/c mice. Replication of the GX18*ca* (**A**) and GX18 (**B**) virus in the respiratory tract of mice. Groups of 6-week-old female BALB/c mice were infected i.n. with 10^6^ EID_50_ of the test viruses. Three mice in each group were killed on days 1, 2, 3, 4, and 5 post inoculation (p.i.) and their nasal turbinates and lungs collected for virus titration in eggs. Virus titers are expressed as the mean ± S.D. log_10_ EID_50_/mL. Samples in which the virus was not detected in 0.1 mL of organ homogenate were assigned the numeric value of 0.5 for calculation purposes. (**C**) Weight changes of the five infected mice from day 0 to day 14 p.i.

**Figure 3 vaccines-13-00596-f003:**
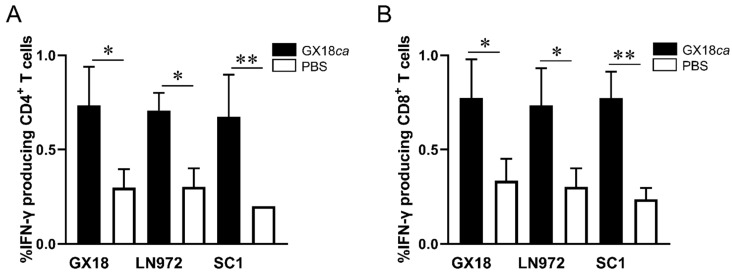
CD4^+^ and CD8^+^ T-cell responses to different influenza viruses in GX18*ca*-inoculated mice. Five 6-week-old female BALB/c mice per group were infected i.n. with 10^6^ EID_50_ of GX18*ca* or PBS (control). At 7 days p.i., samples were collected to test the proportions of IFN-γ-producing CD4^+^ T cells (**A**) and IFN-γ-producing CD8^+^ T cells (**B**) against different antigens, as described in the Materials and Methods Section. Data are shown as mean ± S.D. for each group. Student’s *t*-test was used to compare each group with the PBS-inoculated group (* *p* < 0.05; ** *p* < 0.01).

**Figure 4 vaccines-13-00596-f004:**
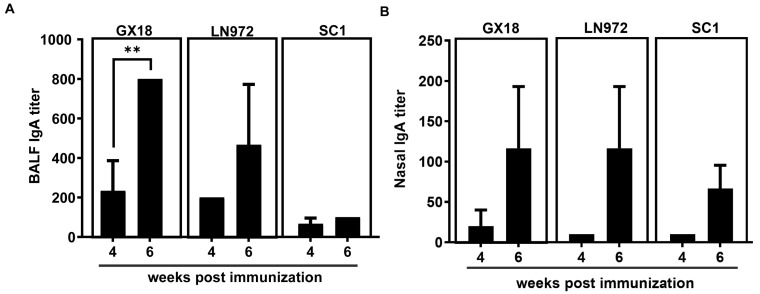
IgA antibody response to homologous and heterologous influenza viruses induced by the GX18*ca* virus in mice. Mice (*n* = 3) immunized i.n. twice with 10^6^ EID_50_ GX18*ca* strain with a 3-week interval. Mice were euthanized 4 and 6 weeks post initial immunization. Bronchoalveolar lavage fluids (**A**) and nasal wash fluids (**B**) were then collected. An enzyme-linked immunosorbent assay (ELISA) was performed to measure IgA antibody levels. The ELISA plates were coated with purified inactivated GX18, LN972, and SC1 antigens. The IgA titers are reported as the mean values of the three samples ± S.D. Statistical significance was determined by Student’s *t* test (** *p* < 0.01).

**Figure 5 vaccines-13-00596-f005:**
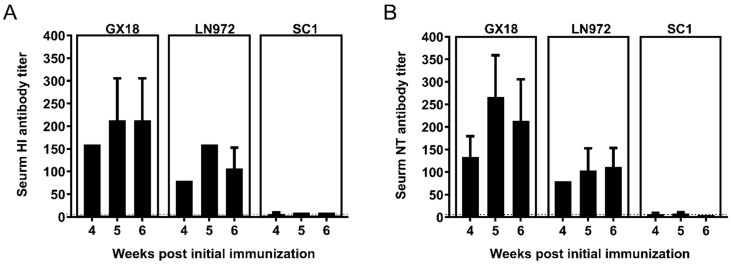
Antibody response to homologous and heterologous influenza viruses induced by the GX18*ca* virus in mice. Groups of thirty mice were immunized with two doses (with a 3-week interval) of 10^6^ EID_50_ of the GX18*ca* vaccine candidate. At 4, 5, and 6 weeks post the initial immunization, three mice were randomly selected from each group for blood collection and serum separation. HI (**A**) and NT (**B**) antibody titers were detected using the homologous GX18 virus and the heterologous LN972 and SC1 viruses, respectively. Negative results were assigned the numeric value of 5 for calculation purposes. The dashed line indicates the limit of detection.

**Figure 6 vaccines-13-00596-f006:**
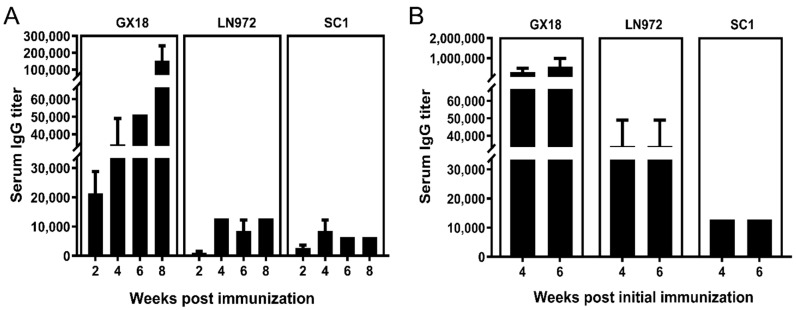
The serum IgG antibody titers in immunized mice at different time points after immunization. Serum samples were collected from the single-immunization group (*n* = 3) at 2, 4, 6, and 8 weeks post immunization (**A**), and from the double-immunization group (*n* = 3) at 4 and 6 weeks post the initial immunization (**B**). The IgG antibody levels in these samples were measured by the ELISA method. Each reported antibody titer is the mean value from three independent mouse samples. Error bars indicate ± S.D.

**Figure 7 vaccines-13-00596-f007:**
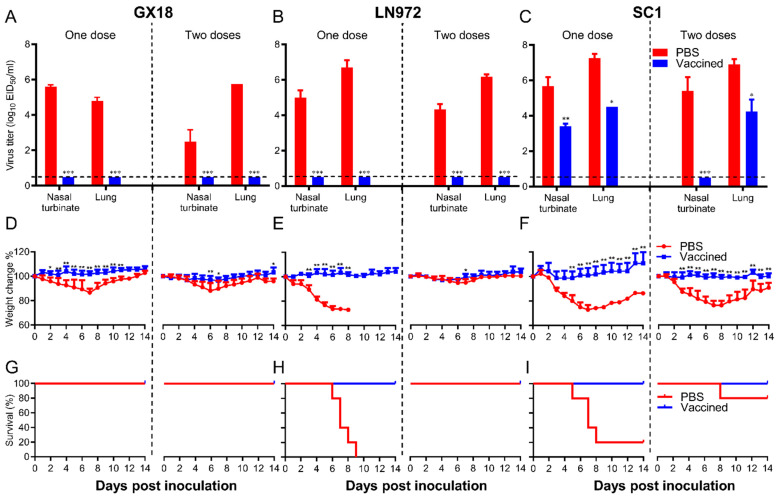
Protective efficacy of GX18*ca* in mice against the GX18, LN972, and SC1 challenge. Groups of eight mice were immunized i.n. with 10^6^ EID_50_ of GX18*ca* once or twice (three weeks apart); three weeks later, mice were anesthetized using a CO_2_ chamber and subsequently challenged i.n. with the indicated viruses. CO_2_ anesthesia is an uncommon procedure, and due to the additional stress it may impose on mice, it could have influenced the survival outcome of the PBS control group. Nasal turbinates and lungs were collected from three mice in each group on day 3 post-challenge for virus titration (**A**–**C**). The viral titers were determined by endpoint titration in eggs and expressed as mean log_10_ EID_50_/mL ± S.D. The body weight (**D**–**F**) and survival (**G**–**I**) of the remaining five mice were monitored for 14 days. Data shown are the mean virus titers of three mice (**A**–**C**) or mean body weights of five mice (**D**–**F**); the error bar shows the standard deviation. The statistical significance of the difference in mean viral titers between immunized mice and mock-immunized controls was analyzed using Student’s *t* test (**A**–**C**, * *p* < 0.05, ** *p* < 0.01, *** *p* < 0.001). To assess the weight change data (presented in panels **D**–**F**), a two-way ANOVA was conducted, with * *p* < 0.05 and ** *p* < 0.01 denoting significant results.

## Data Availability

Data is contained within the article.

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
