# Peer review of "Intranasal Administration of Cold-Adapted Live-Attenuated Eurasian Avian-like H1N1 Vaccine Candidate Confers Protection Against Different-Lineage H1N1 Viruses in Mice"

_vaccines, 2025, doi:10.3390/vaccines13060596_

Round 1
Reviewer 1 Report
Comments and Suggestions for Authors
The authors developed a cold-adapted, temperature-sensitive live-attenuated influenza vaccine (GX18ca) from an EA H1N1 swine virus, showing strong cross-reactive immune responses and full protection in mice.
My main comments are:
The authors claim the Gx18ca is temperature sensitive (Fig. 1). How relevant is the reduced replication at 39 °C? This is outside the physiological temperature range for mice. I would agree if the virus were sensitive to replication in the 35- 37 °C range, but not at 39 °C.
How stable is the cold adaptation? Have you passed the Gx18ca virus (after being cold-adapted) at 37°C to see if the virus will revert to normal growth conditions?
Author Response
- The authors claim the Gx18ca is temperature sensitive (Fig. 1). How relevant is the reduced replication at 39 °C? This is outside the physiological temperature range for mice. I would agree if the virus were sensitive to replication in the 35- 37 °C range, but not at 39 °C.
Answer: Thank you for your thoughtful comments and valuable suggestions regarding our study. We understand your concerns about the temperature sensitivity of the GX18ca vaccine candidate. To address your query, we have conducted extensive experiments to assess the temperature sensitivity phenotype of the virus under various conditions.
Regarding the significant reduction in replication at 39°C, our results clearly show that the GX18ca virus exhibits a marked decrease in replication capacity at this temperature, which confirms its temperature sensitivity. Temperature sensitivity (ts) is a well-recognized characteristic of cold-adapted vaccine candidates and plays an important role in ensuring the safety of live-attenuated vaccines (Chan et al., 2008; Faizuloev et al., 2023). The virus's reduced replication ability at elevated temperatures minimizes potential adverse effects and pathogenic risks. Although 39°C is above the physiological temperature range of mice, it simulates the behavior of the virus at higher temperatures, which is crucial for evaluating its temperature sensitivity phenotype.
Moreover, it is important to note that the normal body temperature of pigs ranges from 38°C to 40°C, with the temperature of the lower respiratory tract often exceeding 39°C. Therefore, replication in the upper respiratory tract (33-35°C) under cold-adapted conditions can elicit strong mucosal immune responses, while preventing replication in deeper, hotter regions of the respiratory tract, which are typically where more pathogenic strains tend to establish. This mechanism reduces the risk of pathogenicity and ensures broad immunity.
Reference:
Chan, W., Zhou, H., Kemble, G., Jin, H., 2008. The cold adapted and temperature sensitive influenza A/Ann Arbor/6/60 virus, the master donor virus for live attenuated influenza vaccines, has multiple defects in replication at the restrictive temperature. Virology 380, 304-311.
Faizuloev, E., Gracheva, A., Korchevaya, E., Smirnova, D., Samoilikov, R., Pankratov, A., Trunova, G., Khokhlova, V., Ammour, Y., Petrusha, O., Poromov, A., Leneva, I., Svitich, O., Zverev, V., 2023. Cold-adapted SARS-CoV-2 variants with different temperature sensitivity exhibit an attenuated phenotype and confer protective immunity. Vaccine 41, 892-902.
- How stable is the cold adaptation? Have you passed the Gx18ca virus (after being cold-adapted) at 37°C to see if the virus will revert to normal growth conditions?
Answer: We sincerely appreciate your thoughtful comments and valuable feedback on our study. In response to your query, we did passage the cold-adapted GX18ca virus at 37°C to assess whether it would revert to normal growth conditions. After passages, the GX18ca virus maintained its cold-adapted (ca) and temperature-sensitive (ts) phenotypes and did not mutate to return to normal growth conditions (related study, unpublished data).
We thank you for your careful review of our work. We hope this explanation clarifies the stability of cold adaptation and the significance of our results.
Reviewer 2 Report
Comments and Suggestions for Authors
The objective of this article is clear, and it is nicely written. I have a few concerns/suggestions to improve the article:
- One of the main concerns I have is about the consistency of PBS vaccinated controls in challenge experiments. Why PBS mice in LN972 two-doses all survived and did not even get sick compared to the PBS mice on one dose group which all died within 9 days? Both are the same PBS group, and this inconsistency raises concern over the methods of infection and repeatability of study. The same happened when infected with SC1 group, PBS groups in one vs two doses and completely different outcomes. This shows that there is inconsistency in infection models which raises concerns over the reliability of other results as well.
- Include ‘mouse model’ in the title.
- Include citations in lines 47-49.
- How much related are the challenge viruses compared to the vaccine virus?
- Line 110, lightly anesthetized, using which anesthesia?
- I would suggest having figures depicting experimental design.
- How many data points are in figure 1? There are no statistical differences which might be due to the lack of replicates.
- Indicate number of replicates or number of animals in each figure legend.
- Figure 2: I would recommend having figures of lung nasal turbinate in A and lungs in B and comparing statistically between GX18 and ca.
- Figure 4 does not show any significance, but text describes about having significant differences.
- What is wpi antigen?
- Figure 7: Add p values using repeated measures ANOVA for body weight change.
Author Response
- One of the main concerns I have is about the consistency of PBS vaccinated controls in challenge experiments. Why PBS mice in LN972 two-doses all survived and did not even get sick compared to the PBS mice on one dose group which all died within 9 days? Both are the same PBS group, and this inconsistency raises concern over the methods of infection and repeatability of study. The same happened when infected with SC1 group, PBS groups in one vs two doses and completely different outcomes. This shows that there is inconsistency in infection models which raises concerns over the reliability of other results as well.
Answer: Thank you very much for your professional review. Regarding the observed inconsistency in outcomes between the PBS control groups in the single-dose and two-dose challenge experiments, we believe this reflects age-related differences in immune development rather than inconsistencies in experimental procedures. Literature indicates that in the single-dose group (inoculated at 6 weeks of age), the immune system is highly reactive (Th1/Th17 dominant) but with insufficient regulation (low Treg activity), leading to higher viral replication peaks and increased susceptibility to excessive inflammation. In contrast, in the two-dose group (inoculated at 9 weeks of age), enhanced immune regulation allows for slower viral clearance but better control of inflammation, leading to higher survival rates and reduced weight loss (Lu et al., 2018).
The increased sensitivity in the single-dose (6-week-old) group more clearly highlights the protective effects of the vaccine, whereas the survival rate in the two-dose (9-week-old) group reflects naturally occurring age-related resistance. Importantly, all vaccine groups (GX18ca) demonstrated consistent protection across both age conditions, confirming that vaccine efficacy is not affected by age-related immune differences. We added “Notably, in the post-immunization challenge test, when exposed to the same viral dose, mice in the double immunized PBS control group exhibited less severe pathogenicity, as evidenced by reduced body weight loss and mortality rates, compared to those with single immunization. This phenomenon is speculated to result from the increased age of the mice.” In discussion.
We sincerely appreciate your insights, and we hope this explanation resolves any concerns regarding the reliability of our infection models and the consistency of our results.
Reference:
Lu, J., Duan, X., Zhao, W., Wang, J., Wang, H., Zhou, K., Fang, M., 2018. Aged Mice are More Resistant to Influenza Virus Infection due to Reduced Inflammation and Lung Pathology. Aging Dis 9, 358-373.
- Include ‘mouse model’ in the title.
Answer: Thank you for your valuable suggestion. We have updated the title to: "Intranasal Administration of Cold-Adapted Live-Attenuated Eurasian Avian-Like H1N1 Vaccine Candidate Confers Protection Against Different-lineage H1N1 Viruses in a Mice".
We appreciate your input, which has helped improve the clarity and specificity of the title.
- Include citations in lines 47-49.
Answer: Thank you very much for your professional review. We have already moved References 4-6 to the end of the sentence to support this view. We appreciate your suggestion, and we hope this revision improves the accuracy and completeness of the manuscript.
- How much related are the challenge viruses compared to the vaccine virus?
Answer: Thank you for your professional and constructive review. Based on a phylogenetic analysis of the HA gene sequences of H1N1 swine influenza viruses (SIVs), the strains used in our study fall into two major evolutionary branches: the Eurasian avian-like H1N1 (EA H1N1) and the classical swine H1N1 (CS H1N1). As shown in the figure, the vaccine candidate strain GX18ca is located within the EA H1N1 SIV branch, along with the wild-type strains SW/GX/18/11 and SW/LN/972/16.
Further analysis reveals the following:
- Within the EA H1N1 branch, GX18ca shares the closest genetic relationship with the parental strain SW/GX/18/11.
- Although SW/LN/972/16 belongs to the EA H1N1 branch, it is genetically more distant from GX18ca.
- The human-origin strain SC/1/09 (H1N1/2009) forms an independent branch within the CS H1N1 lineage and is genetically the most distant from GX18ca.
We hope this explanation clarifies the degree of genetic relatedness between the challenge viruses and the vaccine virus.
These results have been reflected in our previous studies , and we have added the corresponding references to the article. We appreciate your insightful feedback, which has helped improve the manuscript.

Fig. R1. Phylogenetic tree of HA gene
- Line 110, lightly anesthetized, using which anesthesia?
Answer: Thank you for your careful review and insightful question. In this study, mice were lightly anesthetized with carbon dioxide inhalation. We have already made the revisions in the article.
- I would suggest having figures depicting experimental design.
Answer: Thank you for your valuable review and suggestion. For the research method of "2.6 Evaluation of immunogenicity of the GX18ca vaccine candidate in mice", we have rechecked the time points and the number of animals described in the article for detecting the cellular immune responses, humoral immune level, and mucosal immune level to ensure that readers can understand the process of the experiment.
- How many data points are in figure 1? There are no statistical differences which might be due to the lack of replicates.
Answer: Thank you for your insightful comment and for pointing this out. In Figure 1, we included data from three independent biological replicates per group. Additionally, we have performed a statistical analysis to assess the differences between the groups. The updated analysis and results are included in the revised manuscript. We appreciate your thoughtful feedback, which has helped improve the clarity and rigor of our work.
- Indicate number of replicates or number of animals in each figure legend.
Answer: Thank you for your valuable suggestion. We have now included the number of replicates and the number of animals in each figure legend to provide clearer information about the experimental setup. We appreciate your thoughtful feedback, which has helped enhance the clarity of our manuscript.
- Figure 2: I would recommend having figures of lung nasal turbinate in A and lungs in B and comparing statistically between GX18 and ca.
Answer: Thank you for your valuable suggestion. We have respectively presented the dynamic changes of the viruses after infection with GX18ca and GX18 in mice. The purpose is not only to show the replication titers of the GX18ca virus in the turbinate and the lung, but also to more clearly illustrate that GX18ca has a shorter replication time.
- Figure 4 does not show any significance, but text describes about having significant differences.
Answer: Thank you for your careful review. We have performed statistical analysis using unpaired Student’s t-test to assess the SIgA antibody levels in mice from different immunization groups at various time points.
- What is wpi antigen?
Answer: Thank you very much for your professional review. "WPI" stands for weeks post-immunization, and the antigens used in the study refer to the viral strains GX18, LN972, and SC1. We have updated the figure legend to clarify this, and we apologize for any confusion caused by the previous wording. We appreciate your attention to detail, which has helped improve the clarity of our manuscript.
- Figure 7: Add p values using repeated measures ANOVA for body weight change.
Answer: Thank you for your careful review. In response to your suggestion, we have performed statistical analysis using repeated measures ANOVA on the time-series data of body weight changes in mice from different immunization groups post infection. This analysis was conducted to determine if there were significant differences in body weight changes between the groups. We have now added the corresponding p-values in Figure 7 to provide a more complete and rigorous presentation of the results. We appreciate your insightful feedback, which has contributed to improving the statistical clarity and robustness of the manuscript.
Reviewer 3 Report
Comments and Suggestions for Authors
Manuscript: Intranasal Administration of Cold-Adapted Live-Attenuated Eurasian Avian-Like H1N1 Vaccine Candidate Confers Protection Against Different-lineage H1N1 Viruses
The authors have developed a cold-adapted, temperature-sensitive live-attenuated influenza vaccine (GX18ca) which demonstrates broad cross-reactivity and strong immune responses elicited in the mouse model. The authors show that GX18ca induces mucosal and systemic immunity, T-cell responses, and effective protection against both homologous and heterologous viral challenges. However, a few questions arise which need to be addressed:
- Line 193 “Significantly, no virus was isolated from the lungs or nasal turbinates of any mice on day 5 p.i. (Fig. 2B)”. However, Fig. 2B shows the viral titers in the lungs and nasal turbinates of the mice. Did the authors monitor the mice for additional days to determine when the lungs and nasal turbinates became virus-free.
- The authors can also compare the cellular immune efficacy of the GX18ca vaccine candidate, with GX18 vaccine to get a better picture of GX18ca CD4+ T cell responses in comparison to GX18 vaccine. This will show whether GX18ca is a better vaccine or an alternative to GX18.
- The authors have analyzed the cellular immune efficacy of the GX18ca vaccine candidate in the spleen, while they could also check for the same in the lungs of mice. This could also allow them to analyze resident T cell, in addition to other cell populations.
- The authors should also perform virus neutralization assay using serum to get better idea of functional antibodies in the serum.
- The IgG titer for GX18 should be presented as an endpoint titer using serial dilutions to provide a clearer assessment of serum efficacy. At higher concentrations, antibody levels appear uniformly high, making it difficult to distinguish between groups, as seen in Figure 6B.
- In addition to reduced viral titers and clearance by day 5, were there any additional markers like histopathology used to confirm the safety and attenuation of GX18ca?
Author Response
- Line 193 “Significantly, no virus was isolated from the lungs or nasal turbinates of any mice on day 5 p.i. (Fig. 2B)”. However, Fig. 2B shows the viral titers in the lungs and nasal turbinates of the mice. Did the authors monitor the mice for additional days to determine when the lungs and nasal turbinates became virus-free.
Answer: Thank you for your valuable comment. In response, we would like to clarify that we only measured the viral titers in the nasal turbinates and lungs of the mice from days 0 to 5 post-infection with the GX18ca strain. The results showed that between days 1 to 4, the average viral titers in the nasal turbinates ranged from 0.67 to 2.67 log10 EID50/mL, and in the lungs, titers ranged from 1 to 2.5 log10 EID50/mL. By day 5, viral titers had decreased to baseline levels, where no virus was detected. The baseline is defined as the threshold where no viral replication is observed during egg culture, and when virus is undetectable in 0.1 mL of tissue homogenate, it is assigned a value of 0.5 log10 EID50/mL (the "0" on the graph corresponds to this value) for logarithmic calculations and graphical representation. This baseline reflects the sensitivity of our experimental method.
The viral titers continuously decreased from days 1 to 5, and additionally, the mice infected with the GX18ca virus showed a growth trend similar to the PBS control group. Based on these observations, we did not extend the monitoring of viral titers beyond day 5.
We hope this explanation clears up any confusion. Thank you again for your thorough review and insightful questions.
- The authors can also compare the cellular immune efficacy of the GX18ca vaccine candidate, with GX18 vaccine to get a better picture of GX18ca CD4+ T cell responses in comparison to GX18 vaccine. This will show whether GX18ca is a better vaccine or an alternative to GX18.
Answer: Thank you for your valuable feedback and insightful suggestion. We have carefully considered your recommendation to compare the cellular immune efficacy of the GX18ca vaccine candidate with the GX18 vaccine. In this study, GX18ca was developed as a cold-adapted, temperature-sensitive live-attenuated vaccine candidate through the serial passage of the wild-type GX18 strain in eggs at progressively lower temperatures. Our main focus was on evaluating the advantages of GX18ca as a vaccine candidate compared to traditional inactivated vaccines, which is why we did not directly compare GX18 with GX18ca in the experimental design.
However, we recognize that such a comparison could provide a more comprehensive assessment of GX18ca's immune advantages. Theoretically, GX18ca, as a live-attenuated vaccine, can more effectively mimic natural infection, thus triggering a more complete immune response, including both cellular and humoral immunity. While GX18 also elicits immune responses, its pathogenicity limits its use as a vaccine and prevents direct comparison.
Moreover, we have observed that GX18ca induces a strong cellular immune response in mice, particularly involving CD4+ and CD8+ T cells, which are crucial in preventing influenza virus infections. In contrast, GX18 infection may induce similar immune responses, but due to its pathogenicity, it could lead to over activation of the immune system and potential tissue damage.
To directly address your suggestion, we plan to design dedicated experiments in future studies to compare the cellular immune effects of GX18ca and GX18. This will allow us to better understand the advantages of GX18ca as a vaccine candidate and provide stronger evidence for its potential application in the field.
We greatly appreciate your thoughtful suggestion, and we look forward to incorporating these comparisons into our future research.
- The authors have analyzed the cellular immune efficacy of the GX18ca vaccine candidate in the spleen, while they could also check for the same in the lungs of mice. This could also allow them to analyze resident T cell, in addition to other cell populations.
Answer: Thank you very much for your professional review and valuable suggestion. The primary goal of this study was to evaluate the systemic cellular immune responses induced by the GX18ca vaccine in mice to determine whether it could activate the body’s specific cellular immune responses. The spleen, as a critical immune organ rich in immune cells, particularly T cells and B cells, was chosen as the primary site for assessing systemic cellular immunity. It is an ideal location for studying the immune responses induced by the vaccine.
During the experimental design phase, we aimed to collect data at multiple time points to monitor the dynamic changes in the immune response. Due to constraints in experimental resources and the number of mice available, we prioritized spleen sampling to obtain critical immunological data. Additionally, while the lungs are a relevant site for immune responses, given their complexity and the diversity of cell populations, we decided to focus on the spleen for this study.
We do appreciate your suggestion to also analyze the lungs, which could indeed provide valuable insights into resident T cells and other cell populations. This is an excellent idea, and we plan to consider lung tissue analysis in future studies to further expand our understanding of the immune responses induced by GX18ca.
We greatly appreciate your thoughtful feedback and look forward to incorporating this into our future research.
- The authors should also perform virus neutralization assay using serum to get better idea of functional antibodies in the serum.
Answer: Thank you very much for your professional review and valuable suggestion. In response, we have indeed assessed the neutralizing antibodies (NT) in serum by performing a virus neutralization assay. Figure 5B displays the neutralizing antibody titers in the serum of immunized mice. After immunization, the mice produced relatively high levels of hemagglutination inhibition (HI) and neutralizing (NT) antibodies against the wild-type parental virus GX18. These antibody titers peaked in the second week following the second immunization, with HI antibodies reaching 1:213.3 and NT antibodies reaching 1:160.
For the LN972 strain, the HI and NT antibody titers were lower than those induced by GX18 vaccination. Specifically, HI antibodies peaked at 1:160 in the second week following double immunization, while NT antibodies peaked at 1:111.7 in the third week. However, the SC1 (2009/H1N1) strain exhibited the lowest HI and NT antibody titers among the three viruses. These titers remained below 1:50 throughout the three-week period following double immunization.
Overall, these results indicate that the GX18ca vaccine candidate can induce high titers of HI antibodies and neutralizing antibodies against the EA H1N1 viruses, but the titers against the 2009/H1N1 virus are relatively low.
We appreciate your suggestion to perform the neutralization assay, which has provided valuable insight into the functional antibody responses induced by the vaccine.
- The IgG titer for GX18 should be presented as an endpoint titer using serial dilutions to provide a clearer assessment of serum efficacy. At higher concentrations, antibody levels appear uniformly high, making it difficult to distinguish between groups, as seen in Figure 6B.
Answer: Thank you very much for your professional review and valuable suggestions. We are aware of your concern regarding Figure 6B, where at higher concentrations, antibody levels seem uniformly high, making it difficult to differentiate between groups. In the revised version, to tackle this problem, we report the IgG titers for GX18 as endpoint titers via serial dilutions. This approach offers a clearer evaluation of serum efficacy and enables better discrimination among the groups.
- In addition to reduced viral titers and clearance by day 5, were there any additional markers like histopathology used to confirm the safety and attenuation of GX18ca?
Answer: The histopathological examination results showed that mice infected with the wild-type GX18 virus exhibited significant lung pathology, with about 50-70% of the lung tissue affected by moderate hemorrhagic pneumonia by day 3 post-infection. On the contrary, mice inoculated with GX18ca had normal lung tissue with no significant pathological changes on day 3 post-inoculation. Also, no pathological changes were observed in the lungs of the PBS control group mice. These histopathological findings further support the attenuation and safety of GX18ca, as the GX18ca did not induce significant lung pathology, unlike the wild-type virus. Regrettably, the pathological images failed to meet the publication standards, and hence could not be included. We acknowledge this shortcoming and are committed to addressing it in future related experiments. Your valuable feedback has been instrumental in enhancing our understanding and analysis of GX18ca's attenuation, for which we are truly grateful.
Round 2
Reviewer 1 Report
Comments and Suggestions for Authors
None
Author Response
We have already answered the question in Round 1.
Reviewer 2 Report
Comments and Suggestions for Authors
I would like to thank the authors for their responses to my prior questions. However, I could not agree with the justification that the PBS control group has such a remarkable differences across the studies just because of a difference in age 6 week versus 9 week. As a researcher working on influenza and aging, using both humans and animal models, 2-3 month of mouse age reflects adulthood (20+ years), and it is not true that 3 week age differences makes such a huge difference. If the authors also look at the sole article they cited, which says aging is associated with resistance to influenza, the aged mice are of age more than 16 months. 16+ months mice are a model for human aging (65+ years), and that is where your explanation of age-effect holds true. This is a major concern to me and your explanation, unfortunately, is not convincing and it still raises concern for your experimental approach and issues therein for me. Likewise, using CO2 as an anesthesia for intranasal inoculation of influenza in mice also raises concern for me. CO2 is used for euthanasia and terminal procedure and not for light anesthesia. For intranasal inoculation the common practice is isoflurane or ketamine/xylazine anesthesia in mice. Therefore, it appears that there is some flaws in the research procedure and consistency.
Author Response
We fully acknowledge the concerns raised by Reviewer 2 regarding the use of COâ‚‚ anesthesia and its potential impact on animal stress and survival. In response to this important point, we have carefully followed your suggestion. In the revised manuscript, we have added the following clarification both immediately after the description of COâ‚‚ anesthesia in the Materials and Methods section and in the caption of Figure 7: “COâ‚‚ anesthesia is an uncommon procedure and, due to the additional stress it may impose on mice, it could have influenced the survival outcome of the PBS control group.”
We believe that this addition not only addresses the reviewer’s concerns but also enhances the transparency and rigor of our study. We have double-checked these changes to ensure consistency and accuracy throughout the manuscript.
Reviewer 3 Report
Comments and Suggestions for Authors
The manuscript has greatly improved in the newer version. The authors have addressed the reviewers queries and the insights provided will guide the authors for their future studies.
Author Response

(The authors gave the same response as above.)
